# Outreach screening to address demographic and economic barriers to diabetic retinopathy care in rural China

**Baixiang Xiao**[1]**, Gareth D. Mercer**[2]**, Ling Jin**[3]**, Han Lin Lee**[4]**, Tingting Chen**[5]**, Yanfang Wang**[3]**, Yuanping Liu**[3]**, Alastair K. Denniston**[6]**, Catherine A. Egan**[7]**, Jia Li**[8]**, Qing Lu**[8]**, Ping Xu**[8]**, Nathan Congdon**[3,4,8]*

1 Affiliated Eye Hospital of Nanchang University, Nanchang City, China, 2 Department of Ophthalmology and Visual Sciences, McGill University, Montréal, Canada, 3 The State Key Laboratory of Ophthalmology, Zhongshan Ophthalmic Center, Sun Yat-Sen University, Guangzhou City, China, 4 Centre for Public Health, Queen's University Belfast, Belfast, United Kingdom, 5 The Ophthalmology Department of the First Affiliated Hospital of Sun Yat-sen University, Guangzhou, China, 6 University Hospitals Birmingham NHS Foundation Trust, Birmingham, United Kingdom, 7 Moorfields Eye Hospital, NHS Foundation Trust, London, United Kingdom, 8 Orbis International, New York, NY, United States of America

* ncongdon1@gmail.com

**Data Availability Statement:** All relevant data are within the paper and its Supporting Information files.

## Abstract

### Importance

Poor access to existing care for diabetic retinopathy (DR) limits effectiveness of proven treatments.

### Objectives

We examined whether outreach screening in rural China improves equity of access.

### Design, setting and participants

We compared prevalence of female sex, age > = 65 years, primary education or below, and requiring referral care for DR between three cohorts with diabetes examined for DR in neighboring areas of Guangdong, China: passive case detection at secondary-level hospitals (n = 193); persons screened during primary-level DR outreach (n = 182); and individuals with newly- or previously-diagnosed diabetes in a population survey (n = 579). The latter reflected the "ideal" reach of a screening program.

### Results

Compared to the population cohort, passive case detection reached fewer women (50·8% vs. 62·3%, p = 0·006), older adults (37·8% vs. 51·3%, p < 0·001), and less-educated persons (39·9% vs. 89·6%, p < 0·001). Outreach screening, compared to passive case detection, improved representation of the elderly (49·5% vs. 37·8%, p = 0·03) and less-educated (70·3% vs. 39·9%, p<0·001). The proportion of women (59.8% vs 62.3%, P>0.300) and persons aged > = 65 years (49.5% vs 51.3%, p = 0.723) in the outreach screening and population cohorts did not differ significantly. Prevalence of requiring referral care for DR was

**Funding:** NC received funding from: Orbis International, Zhongshan Ophthalmic Center, Sun Yat-sen University, Bayer, Ulverscroft Foundation (UK).

**Competing interests:** The authors have declared that no competing interests exist.

significantly higher in the outreach screening cohort (28·0%) than the population (14·0%) and passive case detection cohorts (7·3%, p<0·001 for both).

## Conclusions and relevance

Primary-level outreach screening improves access for the poorly-educated and elderly, and removes gender inequity in access to DR care in this setting, while also identifying more severely-affected patients than case finding in hospital.

## Introduction

Diabetic Retinopathy (DR) is a progressive disorder among people with Diabetes Mellitus (PwDM), in which high levels of blood glucose exert toxic effects on retinal blood vessels. It is the leading cause of avoidable blindness among working-age adults globally [1], and the only cause of blindness that increased in prevalence globally between 1990 and 2015, largely due to increases in low and middle-income countries [2, 3]. Prevention relies on adequate control of blood glucose levels and early detection and treatment of complications (primarily macular edema and neovascular proliferation), which can prevent 95% of severe vision loss [4]. PwDM with or without pre-existing retinopathy should have regular screening examinations, including assessment of best-corrected visual acuity (BCVA) and retinal examination with pharmacologic dilation of the pupils every 1–2 years [5]. Proven health systems strategies are needed to meet the population demands for DR screening, particularly in settings with limited numbers of eye care professionals.

In China, the prevalence of DM among adults has grown by over ten-fold in the last 30 years from 1% in 1980 [6] to 10.9% (95% CI, 10.4–11.5%) in 2013, and there are now has more PwDM living here than in any other country on earth [7, 8]. The disease is more common in urban areas, but patients in rural settings have greater disease-specific morbidity and mortality [9]. In a population-based study of older adults with DM living in rural areas, the overall prevalence of sight-threatening DR (STDR) was 5%. In the same study, those with previously-diagnosed diabetes were even more likely to have STDR, at 13% [10]. Over half of persons with known diabetes in urban areas have never had an eye examination, while in rural areas the figure exceeds two thirds [11] and only 10% of PwDM in rural China with eye disease have ever been diagnosed and treated [12]. In part, this is due to the fact that eye care services are generally only available in secondary-and tertiary-level hospitals, and access to care is poor in rural settings [11]. Furthermore, despite national rates of medical insurance coverage as high as 95% [13], direct and indirect costs are a persistent barrier to accessing health care among rural-dwellers [14]. Inequitable access is a major problem in receiving care for both DM [15] and DR [11], with women, the elderly and the poorly-educated at significant disadvantages.

In rural areas, primary health care is delivered through a combination of township health centres (staffed by general practitioners) and village health posts (staffed by village health workers). In order to improve access to DR care for rural PwDM, the Zhongshan Ophthalmic Center and Orbis International, an eye health non-governmental organization active in China, established in 2017 an outreach DR screening program at township health centres in counties in Guangdong province with a population of approximately 37 million. The program offers free eye examinations and patient education on the importance of regular eye care for all PwDM currently registered at the health centres.

In the current paper, we compare representation of traditionally-underserved groups (women, those aged $\geq$ = 65 years and persons with primary education or below) in this primary-level outreach screening program with a cohort detected in a neighboring area under the current standard model of passive case finding at secondary-level hospitals [16]. We further compared both cohorts to PwDM identified in a recent population-based study [10] in the area, as a reflection of the "ideal" reach of a screening outreach program. Our study hypothesis was that representation of underserved groups would be improved in the outreach screening as compared to passive case-finding cohorts, and the former would more closely resemble the proportion in the population.

## Methods

### Study setting and target population

The cohorts analysed in this study were recruited in rural Guangdong province, China, (per capita gross domestic product [GDP]: US$12,125 in 2019) between 2014 and 2019, and included adults aged 50 years and over with previously- or newly-diagnosed diabetes mellitus (DM). Approval for the parent studies from which the secondary-level [16] and population [10] cohorts were drawn, and for enrolment of new patients from the township clinics, were all provided by the Ethics Committee at the Zhongshan Ophthalmic Centre, Sun Yat-sen University, Guangzhou, China, written informed consent was obtained from all participants, and the tenets of the Declaration of Helsinki were followed throughout.

### Study design

We compared cross-sectional data from three cohorts of people with DM: those presenting spontaneously for eye examinations at five secondary-level hospitals (passive detection cohort); those screened through a primary-level DR outreach program at a single Township Health Centre (outreach screening cohort); and those examined as part of a population-based survey in a single nearby county (population-based cohort). These data were drawn from three separate studies, which were all conducted in rural regions of a single province in China between 2014 and 2019. Each of these studies, and the inclusion criteria used to derive the samples for the present study, are described below.

**Secondary-level passive case-finding cohort.** Patients in the passive detection cohort were drawn from a previously-reported randomized controlled trial investigating a mobile phone reminder system for diabetic retinopathy screening [16]. The study screened 233 consecutive patients presenting between 1 March 2015 and 31 May 2016 for eye examinations at five county hospitals in Guangdong: Shaoguan (population: 2·97 million, 2017 GDP per capita: US $6,250), Chenghai (population: 0·75 million, 2017 GDP per capita: US$8,607), Luoding (population: 1·27 million, 2017 GDP per capita: US$3,134), Huidong (population: 0·93 million, 2017 GDP per capita: US$9,886) and Jieyang (population: 6·09 million, 2017 GDP per capita: US $4,862). Patients either had previously diagnosed DM, or were diagnosed at the time of screening based on characteristic eye findings. We included baseline data on all patients from both the intervention and comparison arms of the trial (n = 233). To enable direct comparison with the population-based cohort, we excluded people younger than 50 years old (40/233 = 17·2%), yielding an analytic sample of 193 individuals.

**Outreach screening cohort.** Members of this cohort were consecutive patients presenting for DR screening at Pingshan Township Health unit in Huidong County (population: 0·93 million, 2017 GDP per capita: US$9,886) from June to September 2019, as part of a primary-level screening program established in 2017. All patients had previously-diagnosed DM based on rapid plasma glucose testing and were registered at the Township Health Centre as required

by Chinese regulations for chronic disease management. All such patients at the Pingshan Health Centre (total n = 937) were eligible, with all those (210 people with diabetes milletus—PwDM) presenting during the study period requested to join the study, 202 were recruited (96.2%). To enable direct comparison with the population-based cohort, we excluded people younger than 50 years old (17/202 = 8·4%), yielding the analytic sample of 182 individuals.

**Population cohort.**  Patients were drawn from the Yangxi Eye Study, a population-based, cross-sectional study involving 6,425 people aged 50 years or older living in Yangxi, South Guangdong Province, between August and November 2014 [10]. Participants were sampled using cluster random sampling, received ocular examinations with dilation of the pupil at home and screened for DM using point of service glycosylated Hb A1c testing (Afinion AS100; Axis-Shield, Norway). For the present analysis we included all participants (n = 579) with HbA1c > 6.5% or who reported a previous physician diagnosis of DM or taking diabetes medications.

## Data collection and variables

We combined the three cohorts into a single dataset and created a categorical variable indicating study group. At the baseline visit of the respective studies, participants all completed similar structured questionnaires with trained research staff. The following indicators of socioeconomic disadvantage were measured in all studies: female gender, age ≥65 years and primary school educational attainment or below. The following additional socioeconomic indicators were measured for the passive case-detection and outreach screening cohorts only: occupation, access to health insurance, and a composite measure of household wealth based on ownership of a set 13 durable assets, as described in the China Rural Household Survey Yearbook (Department of Rural Surveys, National Bureau of Statistics of China, 2013). For all three cohorts, presenting visual acuity (PVA) and best-corrected visual acuity (BCVA) were tested using Early Treatment Diabetic Retinopathy Study (ETDRS) charts [17] and methodologies as previously described [10, 16]. Severity of visual impairment was based on presenting visual acuity (PVA) in the better-seeing eye as per the World Health Organization definitions: mild or none (PVA ≥6/18), moderate (PVA <6/18 and ≥6/60), severe (PVA <6/60 and ≥3/60), and blind (PVA <3/60). DR and diabetic macular edema (DME) assessments were performed by trained non-medical graders in the outreach screening and population cohorts, based on two dilated fundus images for each eye, one centered on the macula and the other one on the optic disc. In the passive detection cohort, DR assessment was based on clinical evaluation during dilated fundus examinations as carried out by trained local ophthalmologists. DR was graded in all three cohorts based on the United Kingdom National Diabetic Eye Screening Program guidelines [18]. Sight-threatening diabetic retinopathy (STDR) was defined as severe pre-proliferative (R2) or proliferative (R3) retinopathy with or without diabetic macular edema (M1). Non-gradable fundus photographs (77/954 = 8·0% across the three cohorts) were treated as STDR because they would require referral for specialist evaluation.

## Statistical analysis

Two-way contingency tables were used to compare the distribution of socioeconomic and clinical characteristics between each pair of cohorts. Differences between each pair of cohorts were tested using Pearson's chi-square tests or Fisher's exact tests if the chi-square approximation was judged to be incorrect due to small cell counts. For the three key indicators of socioeconomic disadvantage and for STDR, we calculated the prevalence in each cohort as a ratio of the prevalence observed in the population-based cohort. We estimated 95% confidence intervals using the likelihood scores method for two binomial proportions, and tested pair-wise

differences between cohorts using two-proportion z-tests. We performed sensitivity analyses by re-introducing participants younger than 50 years old in the outreach screening and passive case-finding cohorts and performing the same 3-way comparison of socioeconomic disadvantage and sight-threatening DR. We did not observe any substantive changes in our principal findings (S1 Fig).

## Results

Compared to the population-based cohort, individuals identified through passive case detection were significantly less likely to be female (62.3% vs. 50.8%, p = 0·005), 65 years or older (51.3% vs. 37.8%, p = 0·002), and to have primary school-level education or below (89.6% vs. 39.9%, p < 0·001; Fig 1). However, outreach screening was significantly more likely than passive case detection to reach people aged 65 years and above (49.5% vs. 37.8%, p = 0·03) and individuals with minimal education (70.3% vs. 39.9%, p < 0·001). While those in the outreach screening group were also less likely to have low education levels compared to the population cohort (70.3 vs 89.6%, p<0.001), the proportion of women and older persons identified did not differ significantly. In addition, compared to passive case detection, those in the outreach screening group were more likely to be in the lowest tertile of household wealth, to be peasant farmers and to rely on insurance to pay for medical expenses (Table 1).

The proportion of people needing referral for more detailed retinal examination (sight-threatening diabetic retinopathy–STDR and those with non-gradable eyes) was significantly higher among the outreach screening cohort (28.6%) than both the population-based cohort (14.1%) and the passive case detection cohort (6.7%; Fig 1). This was due to a combination of higher levels of proliferative retinopathy, macular edema and more non-gradable fundus images in this group (Table 2). The passive case detection and outreach screening patients had longer average durations of diabetic disease and were more likely to be using insulin than the

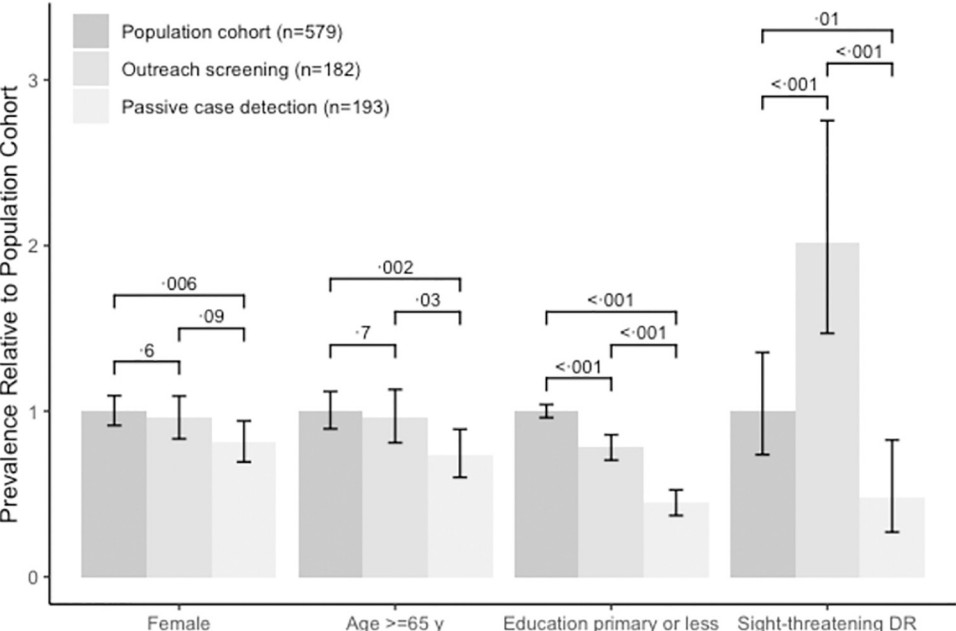

**Fig 1. Comparison of indicators of socioeconomic disadvantage and sight threatening DR between cohorts of patients with DM accessed through passive case detection at secondary level hospitals (n = 193) and primary-level outreach screening (n = 182) with reference to a population-based cohort (n = 579).**

**Table 1. Comparison of additional socioeconomic characteristics of participants in in the passive case detection and outreaching screening cohorts.**

| Characteristics | 1. Passive case detection at secondary level ($n$ = 193) | 2. Outreach screening at primary level ($n$ = 182) | p-value |
|---|---|---|---|
| Occupation, n (%) | | | <0·001[&] |
| Peasant farmer | 59 (30.6) | 100 (54.9) | |
| Other professions | 43 (22.3) | 13 (7.1) | |
| Unemployed/Retired | 91 (47.2) | 68 (37.4) | |
| *Missing* | *0* | *1* | |
| Household asset score tertile, n (%) | | | <0·001[*] |
| Highest | 56 (29.0) | 55 (30.2) | |
| Middle | 84 (43.5) | 21 (11.5) | |
| Lowest | 53 (27.5) | 106 (58.2) | |
| *Missing* | *0* | *2* | |
| Usual mode of transport to hospital, n (%) | | | <0·001[&] |
| Walk | 33 (17.1) | 6 (3.3) | |
| Bicycle | 6 (3.1) | 2 (1.1) | |
| Motorcycle | 41 (21.2) | 49 (27.4) | |
| Public transportation | 82 (42.5) | 66 (36.9) | |
| Car | 31 (16.1) | 56 (31.3) | |
| *Missing* | *0* | *3* | |
| Payment of medical expenses, n (%) | | | <0·001[*] |
| Entirely out of pocket | 83 (43.1) | 18 (9.9) | |
| Full or partial coverage by insurance | 110 (56.9) | 164 (90.1) | |
| *Missing* | *0* | *0* | |

NA = Not applicable

[*]Pearson's Chi-Square

[&]Fisher's Exact Test.

population-based screening cohort, but were similar to each other in these respects. Visual impairment was also significantly more common among the outreach screening than either the population-based and passive detection cohorts.

## Discussion

With an estimated 114 million people living with diabetes [19] and almost half having never had an eye examination [12], there is an imperative in China to improve access to screening for diabetic retinopathy. Nowhere is this truer than in rural areas, where complications from diabetes are more common [9] and where financial and geographic barriers to reaching eye clinics are greater [11]. In theory, health system interventions to improve access to diabetic retinopathy screening may address barriers related to one or more of the following: approachability, acceptability, availability, accommodation, affordability and appropriateness [20]. In the parts of Guangdong covered by the Orbis-ZOC program, primary-level outreach screening is designed to improve approachability through targeted education about the importance of screening. Additionally, by locating services in township health units, screening becomes more available, and by making it free, it becomes more affordable.

In this study we estimated rates of socioeconomic disadvantage among adult PwDM presenting passively to eye clinics in secondary-level hospitals in rural Guangdong province, reflecting the current standard care pathway. Women, older adults and those with lower educational attainment were significantly less likely to engage through the standard pathway of

**Table 2. Clinical characteristics of participants in each setting.**

| Clinical Characteristics | 1. Passive case detection at secondary level (*n* = 193) | 2. Outreach screening at primary level (*n* = 182) | 3. Population-based cohort (*n* = 579) | Global p-value | Pairwise p-values | | |
|---|---|---|---|---|---|---|---|
| | | | | | 1 vs 2 | 1 vs 3 | 2 vs 3 |
| Sight-threatening diabetic retinopathy, n (%) | | | | <0·001* | <0·001* | <0·011* | <0·001* |
| No | 180 (93.2) | 130 (71.4) | 431 (85.9) | | | | |
| Yes | 13 (6.7) | 52 (28.6) | 71 (14.1) | | | | |
| *Missing* | *0* | *0* | *77* | | | | |
| Highest DR grade in either eye, n (%) | | | | <0·001& | <0·001& | <0·001& | <0·001& |
| R0 | 104 (53.9) | 109 (59.9) | 417 (83.5) | | | | |
| R1 | 77 (39.9) | 23 (12.6) | 27 (5.4) | | | | |
| R2 | 12 (6.2) | 15 (8.2) | 8 (1.6) | | | | |
| R3 | 0 (0) | 7 (3.8) | 2 (0.4) | | | | |
| Ungradable | 0 (0) | 28 (15.4) | 49 (9.8) | | | | |
| *Missing* | *0* | *0* | *76* | | | | |
| DME in either eye, n (%) | | | | <0·001* | <0·001* | <0·001& | 0.029* |
| Yes (M1) | 2 (1.0) | 10 (5.5) | 15 (3.0) | | | | |
| No (M0) | 191 (99.0) | 144 (79.1) | 438 (87.3) | | | | |
| Ungradable | 0 (0) | 28 (15.4) | 49 (9.8) | | | | |
| *Missing* | *0* | *0* | *77* | | | | |
| Best-corrected visual acuity, better eye, n (%) | | | | 0.001& | <0·001& | 0·061& | 0·017& |
| Mild or no VI | 186 (96.4) | 153 (84.5) | 515 (91.5) | | | | |
| Moderate VI | 7 (3.6) | 22 (12.2) | 30 (5.3) | | | | |
| Severe VI | 0 (0) | 3 (1.7) | 7 (1.2) | | | | |
| Blind | 0 (0) | 3 (1.7) | 11 (2.0) | | | | |
| *Missing* | *0* | *1* | *16* | | | | |
| Duration of diabetes, n (%) | | | | <0·001& | 0·319& | <0·001& | <0·001& |
| New diagnosis | 0 (0) | 3 (1.7) | 434 (78.6) | | | | |
| <5 years | 72 (37.3) | 56 (31.6) | 76 (13.8) | | | | |
| 5–10 years | 62 (32.1) | 58 (32.8) | 36 (6.5) | | | | |
| 11–20 years | 53 (27.5) | 51 (29.8) | 5 (0.9) | | | | |
| >20 years | 6 (3.1) | 9 (5.3) | 1 (0.2) | | | | |
| *Missing* | *0* | *5* | *27* | | | | |
| Previous diabetes treatment, n (%) | | | | <0·001& | 0·004& | <0·001& | <0·001& |
| Oral Hypoglycemics | 112 (65.1) | 127 (72.2) | 497 (86.9) | | | | |
| Insulin | 34 (19.8) | 38 (21.6) | 4 (0.7) | | | | |
| Lifestyle changes | 8 (4.7) | 0 (0) | 0 (0) | | | | |
| None | 16 (9.3) | 4 (2.3) | 9 (1.6) | | | | |
| Don't know | 2 (1.1) | 1 (0.57) | 1 (0.2) | | | | |
| *Missing* | *21* | *6* | *7* | | | | |
| Previously treated for diabetic eye disease, n (%) | | | | <0·001* | 0·090* | <0·001* | 0·002& |
| Yes | 34 (17.8) | 19 (11.0) | 5 (4.2) | | | | |
| No | 143 (74.9) | 148 (84.1) | 113 (95.7) | | | | |
| Don't know | 14 (7.3) | 9 (5.1) | 0 (0) | | | | |
| *Missing* | *2* | *6* | *0* | | | | |

*VI* = Visual Impairment, Sight-threatening diabetic retinopathy = severe pre-proliferative (R2) or proliferative (R3) retinopathy with or without diabetic macular edema (M1), or ungradable fundus image.

Statistical significance tests

*Pearson's Chi-Square

&Fisher's Exact Test.

care, relative to their proportions in the affected population. A prior study in Guangdong observed lower screening coverage among people with lower educational attainment and lower monthly income, but found no relationship with gender or age [11]. A likely explanation for the discrepant findings is that the prior study combined patients from a tertiary and a secondary level hospital, whereas our study included only participants at the secondary level and below.

We further estimated improvements in health equity achievable through implementation of primary-level outreach screening. We observed increased access along all three dimensions of socioeconomic disadvantage, with the largest benefits being seen among patients with lower education. Previous evidence is scarce on reduction in inequities for care of diabetic retinopathy with outreach screening. A randomized trial in Hong Kong found that providing free screening for DR resulted in moderately improved uptake [21], while a trial among low-income PwDM in New York City reported that an individualized telephone intervention targeting patients' knowledge about diabetic retinopathy and stage of change in managing their diabetes (i.e. approachability and acceptability) increased screening uptake by 74%, compared to a printed information brochure [22]. The effect of the latter intervention did not differ by patient ethnicity or language. Neither study reported intervention effects disaggregated by other indicators of socioeconomic status or other indicators of health equity.

Nevertheless, we observed persistent inequities in the outreach screening sample compared to the population as a whole, particularly for the very elderly (aged older than 75 years). Barriers faced by older rural Chinese to attending primary level clinics include poorer overall health, and lower household incomes [23, 24]. Further research is necessary to determine how these and other factors can be addressed to improve access for this vulnerable group.

With respect to burden of illness, we observed that participants in primary-level outreach screening were significantly more likely to require referral for definitive eye care than those presenting spontaneously to secondary level hospitals or than the population as a whole. A similar phenomenon was observed in the trial of free screening in Hong Kong, in which the intervention group had higher rates of retinopathy [21]. In contrast, a study on glaucoma screening in China demonstrated improvement in equity but subjects identified were less severely affected than those coming to clinic spontaneously [25]. In our study, a potential cause for higher rates of STDR among outreach screening patients than the population-based cohort is that the latter had a far greater proportion of people with newly diagnosed diabetes. However, this does not explain the difference between the outreach screening cohort and passive case detection cohorts, since, for these two cohorts, the distribution of diabetes duration was similar. Another potential explanation is that patients in the outreach screening cohort were less likely to report having previously been treated for diabetic eye disease than passive case detection patients.

This portion of our analysis was limited by differences in outcome ascertainment between the three cohorts. The outreach screening and population-based studies employed two non-medical graders to evaluate standard fundus photographs, with expert adjudication by an ophthalmologist of conflicting results. By contrast, in the passive case detection cohort, DR was ascertained through fundus examinations performed by trained rural ophthalmologists. Our group previously found that rural ophthalmologists in this region detect proliferative DR with 66% sensitivity and 91% specificity, and macular oedema with 65% sensitivity and 95% specificity [26]. Applying standard bias analysis methods for differential misclassification of binary outcomes [27], we estimated that the observed disparity in levels of STDR between outreach screening and passive case detection is likely a conservative estimate of the true disparity (results not shown). Though much of the difference between groups with respect to need for referral care was due to a higher rate of ungradable images in the outreach screening cohort,

the consistent finding of significantly worse vision impairment in this group suggests that real clinical differences were present.

A further limitation of this study is that communities were not randomly assigned to passive case detection and outreach screening, which allows for potential selection bias due to unmeasured area-level variables, such as quality of primary health services and availability of public transportation. The potential impact of these factors demands further study. Additionally, some of the socioeconomic differences observed may be attributable to temporal differences in the period of data collection across the three studies. However, there have not been significant changes, such as modifications to insurance coverage at the national level, over the 5-year period that would be expected to impact on equity of access.

Despite these limitations, our finding that primary-level outreach screening for DR is likely to reduce both the population burden of avoidable visual impairment and its associated social inequity is of great relevance to health policy makers. Not only do socioeconomically-disadvantaged groups tend to experience poorer access to health care, they frequently also carry an inequitable burden of disease [28]. To achieve potential gains, screen-positive patients, including those with incidental findings, must actually receive appropriate management. Previous work in other settings has shown that this is not always the case for vision outreach screening [29]. In Guangdong, the Zhongshan Ophthalmic Center has expertise and capacity to treat these patients, but further work is required to evaluate the integrity of the referral system and whether referred patients actually receive the tertiary care they need.

## Supporting information

**S1 Fig. Prevalence of indicators of socioeconomic disadvantage and sight threatening diabetic retinopathy among groups of people with diabetes detected through primary-level outreaching screening and passive case detection at secondary-level hospital relative to a population-based cohort.**
(TIF)

**S1 Dataset.**
(CSV)

## Acknowledgments

The authors acknowledge the doctors and nurses from the community health centers in Yuexiu District, Guangzhou City, Chenghai District People's Hospital and Shaoguan Hospital for their contribution to data collection for the study.

## Author Contributions

**Conceptualization:** Nathan Congdon.

**Data curation:** Baixiang Xiao, Gareth D. Mercer, Han Lin Lee.

**Formal analysis:** Baixiang Xiao, Gareth D. Mercer.

**Investigation:** Baixiang Xiao, Tingting Chen, Yanfang Wang, Yuanping Liu.

**Methodology:** Nathan Congdon.

**Project administration:** Baixiang Xiao.

**Resources:** Baixiang Xiao.

**Supervision:** Baixiang Xiao, Nathan Congdon.

**Validation:** Ling Jin, Nathan Congdon.

**Writing – original draft:** Baixiang Xiao.

**Writing – review & editing:** Baixiang Xiao, Gareth D. Mercer, Han Lin Lee, Tingting Chen, Alastair K. Denniston, Catherine A. Egan, Jia Li, Qing Lu, Ping Xu, Nathan Congdon.

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
