## [Decision Letter · Decision Letter 0]

18 Dec 2021

PONE-D-21-16668Outreach screening to address demographic and economic barriers to diabetic retinopathy care in rural ChinaPLOS ONE

Dear Dr. Congdon,

Thank you for submitting your manuscript to PLOS ONE. After careful consideration, we feel that it has merit but does not fully meet PLOS ONE’s publication criteria as it currently stands. Therefore, we invite you to submit a revised version of the manuscript that addresses the points raised during the review process.

We look forward to receiving your revised manuscript.

Kind regards,

Soujanya Kaup, MS DNB FPRS

Academic Editor

PLOS ONE

Journal Requirements:

Natural Earth (public domain): http://www.naturalearthdata.com/.

6. Please include your tables as part of your main manuscript and remove the individual files. Please note that supplementary tables (should remain/ be uploaded) as separate "supporting information" files

Additional Editor Comments (if provided):

Dear authors,

The study has important conclusions which can impact Diabetic retinopathy screening programmes. However, as pointed out by the reviewers, this study has major limitations which need strong justification, especially the fact that the the three comparison groups differ in the time period of examinations and also differ geographically.

Please find the reviewer comments below. I have recommended "major revision" for these reasons.

Reviewers' comments:

Reviewer's Responses to Questions

**Comments to the Author**

1. Is the manuscript technically sound, and do the data support the conclusions?

Reviewer #1: Yes

Reviewer #2: Yes

Reviewer #3: No

2. Has the statistical analysis been performed appropriately and rigorously? 

Reviewer #1: Yes

Reviewer #2: Yes

Reviewer #3: Yes

3. Have the authors made all data underlying the findings in their manuscript fully available?

Reviewer #1: No

Reviewer #2: Yes

Reviewer #3: Yes

4. Is the manuscript presented in an intelligible fashion and written in standard English?

Reviewer #1: Yes

Reviewer #2: Yes

Reviewer #3: Yes

5. Review Comments to the Author

Reviewer #1: Here is a list of specific comments. Note: line and page numbering are not available; line and page numbering in reviews and comments is based on those in the Editorial Manager-generated PDF.

1. Another drawback of this manuscript was the lack of health outcome comparisons.

2. Page 13 of 24, Secondary-level passive case-finding: I suggest revising “patients were drawn” as ‘patients in the passive detection cohort were drawn’.

3. Page 13 of 24, Outreach screening: Please include the description of the 185 patients in the outreach screening cohort.

4. Page 14 of 24, 1st paragraph: I suggest relocating the sentence “to create comparable inclusion criteria, . . . ” to the Study Design section.

5. Page 15 of 24, 1st paragraph: Although the goal was to compare the passive detection cohort and the outreach screening cohort to the population-based cohort, it would be necessary to provide results of chi-square tests for the distribution of socioeconomic and clinical characteristics among three cohorts; i.e., add a column of p-value for overall tests in Table 2.

6. Page 15 of 24, 1st paragraph, chi-square tests: Please confirm if Fisher’s exact tests were necessary for some characteristics with 0-count cells such as highest DR grade in either eye, DME in either eye, etc.

Reviewer #2: Abstract

Please indicate if there was a difference in the less educated between the population cohort and outreach cohort. You only mentioned there was no difference in age and women between both cohorts

Methodology

The outreach cohort is clearly cross sectional but the population and secondary level cohort is a bit confusing. Are the population cohort and the secondary level cohort retrospective with regards to this particular study in which the data were collected during the trial (for the secondary cohort) or at some time in the past for the population cohort OR were they cross sectional where all participants were re-invited, screened, and questionnaires administered? Kindly clarify

If cross sectional, did all the initial patients present for the current study? if not what proportion did not?

Discussion

Please could you explain why the outreach screening identifies more severely-affected patients than case finding in hospital. One would think that the patients presenting to the hospital would have worse disease

Reviewer #3: The authors have tried to test their hypothesis by comparing 3 cohorts of patients that were from different studies and done at different period and all in different regions.

These regions may all be rural but in a country like China, rural populations and areas are heterogeneous. The study periods range from 2014 for one cohort to 2019 in the recent cohort. Lots of progress has been made in 5 years and so it is difficult to make these conclusions.

6. PLOS authors have the option to publish the peer review history of their article (what does this mean?). If published, this will include your full peer review and any attached files.

Reviewer #1: No

Reviewer #2: No

Reviewer #3: No

---

## [Author Response · Author response to Decision Letter 0]

19 Jan 2022

Response to Reviewers

Response: Thank you. The letter, marked and clean versions of the revised manuscripts are now attached.

2. Response: We do not have any changes to the finance disclosure.

Response: N/A

Response: The style and format have been changed accordingly.

Response: We are now uploading the related data simultaneously with the manuscript.

Response: The phrase "data not shown" is now deleted. These data are now uploaded as Supplementary Figure 1.

Response: We have made this change.

Natural Earth (public domain): http://www.naturalearthdata.com/.

Response: Thank you for the information. Figure 1 is now deleted.

6. Please include your tables as part of your main manuscript and remove the individual files. Please note that supplementary tables (should remain/ be uploaded) as separate "supporting information" files

Response: Tables are now incorporated in the manuscript.

Additional Editor Comments (if provided):

Dear authors,

The study has important conclusions which can impact Diabetic retinopathy screening programmes. However, as pointed out by the reviewers, this study has major limitations which need strong justification, especially the fact that the three comparison groups differ in the time period of examinations and also differ geographically.

Please find the reviewer comments below. I have recommended "major revision" for these reasons.

Reviewers' comments:

Reviewer's Responses to Questions

Comments to the Author

Review Comments to the Author

Reviewer #1: Here is a list of specific comments. Note: line and page numbering are not available; line and page numbering in reviews and comments is based on those in the Editorial Manager-generated PDF.

1. Another drawback of this manuscript was the lack of health outcome comparisons.

Response: The principal aim of this MS was not to assess health outcomes, but rather to assess the impact of screening outreach strategies on equity of access for diabetic retinopathy care. However, we do compare the % of persons requiring referral care for DR between cohorts.

2. Page 13 of 24, Secondary-level passive case-finding: I suggest revising “patients were drawn” as ‘patients in the passive detection cohort were drawn’.

Response: This has been modified.

3. Page 13 of 24, Outreach screening: Please include the description of the 185 patients in the outreach screening cohort. 

Response: We include a description of this cohort in the section of the methods titled “Outreach screening cohort”. We have clarified the wording of the Study Design section to make it clear that we are describing cohorts drawn from three separate studies.

4. Page 14 of 24, 1st paragraph: I suggest relocating the sentence “to create comparable inclusion criteria, . . . ” to the Study Design section. 

Response: Thank you. As suggested, we have removed this to the Study Design section.

5. Page 15 of 24, 1st paragraph: Although the goal was to compare the passive detection cohort and the outreach screening cohort to the population-based cohort, it would be necessary to provide results of chi-square tests for the distribution of socioeconomic and clinical characteristics among three cohorts; i.e., add a column of p-value for overall tests in Table 2.

Response: Thank you. We have included p-values from global significance testing (Chi Square or Fisher exact tests) in Table 2.

6. Page 15 of 24, 1st paragraph, chi-square tests: Please confirm if Fisher’s exact tests were necessary for some characteristics with 0-count cells such as highest DR grade in either eye, DME in either eye, etc. 

Response: We did use Fisher’ exact test under these circumstances, as described in the Statistical Methods section.

Reviewer #2: Abstract

Please indicate if there was a difference in the less educated between the population cohort and outreach cohort. You only mentioned there was no difference in age and women between both cohorts

Response: Individuals with lower educational attainment were significantly better represented in the population-based cohort than in either of the other two cohorts, as well as in the primary-level outreach screening cohort than in the passive case detection cohort. We describe these findings in the first paragraph of the results section and in Figure 1.

Methodology

The outreach cohort is clearly cross sectional but the population and secondary level cohort is a bit confusing. Are the population cohort and the secondary level cohort retrospective with regards to this particular study in which the data were collected during the trial (for the secondary cohort) or at some time in the past for the population cohort OR were they cross sectional where all participants were re-invited, screened, and questionnaires administered? Kindly clarify. If cross sectional, did all the initial patients present for the current study? if not what proportion did not?

Response: Thank you for this point. All three cohorts used cross-sectional data. The passive case detection cohort, though drawn from a longitudinal RCT, used only baseline data from the original study. No participants were re-invited. The text has been clarified on this point.

Discussion

Please could you explain why the outreach screening identifies more severely-affected patients than case finding in hospital. One would think that the patients presenting to the hospital would have worse disease 

Response: As mentioned in the Discussion section, our hypothesis is that disadvantaged people with severe disease are prevented by barriers of age and economic circumstance from coming to hospital; only outreach screening will serve them.

Reviewer #3: The authors have tried to test their hypothesis by comparing 3 cohorts of patients that were from different studies and done at different period and all in different regions.

These regions may all be rural but in a country like China, rural populations and areas are heterogeneous. The study periods range from 2014 for one cohort to 2019 in the recent cohort. Lots of progress has been made in 5 years and so it is difficult to make these conclusions. 

Response: These areas are all rural regions within a single province in China. While medical progress has been relatively quick in China, there have not been significant changes, such as modifications to insurance coverage at the national level, over this 5-year period that would be expected to impact on equity of access. Nonetheless, this temporal difference between cohorts has been added to the limitations section.

6. PLOS authors have the option to publish the peer review history of their article (what does this mean?). If published, this will include your full peer review and any attached files.

Do you want your identity to be public for this peer review? For information about this choice, including consent withdrawal, please see our Privacy Policy.

Reviewer #1: No

Reviewer #2: No

Reviewer #3: No

---

## [Editor Report · Decision Letter 1]

21 Mar 2022

Outreach screening to address demographic and economic barriers to diabetic retinopathy care in rural China

PONE-D-21-16668R1

Dear Dr. Congdon,

We’re pleased to inform you that your manuscript has been judged scientifically suitable for publication and will be formally accepted for publication once it meets all outstanding technical requirements.

Kind regards,

Soujanya Kaup, MS DNB FPRS

---

## [Editor Report · Acceptance letter]

6 Apr 2022

PONE-D-21-16668R1 

Outreach screening to address demographic and economic barriers to diabetic retinopathy care in rural China 

Dear Dr. Congdon:

I'm pleased to inform you that your manuscript has been deemed suitable for publication in PLOS ONE. Congratulations! Your manuscript is now with our production department. 

Kind regards, 

on behalf of

Dr. Soujanya Kaup 

Academic Editor

PLOS ONE